# Enhanced performance of mixed HWMA-CUSUM charts using auxiliary information

**Faiza Zubair** [1] *, **Rehan Ahmad Khan Sherwani**[2], **Muhammad Abid**[3]

**1** Higher Education Department, Lahore, Pakistan, **2** University of the Punjab, Lahore, Pakistan, **3** Government College University, Faisalabad, Pakistan

* faizadaud1986@gmail.com

## Abstract

Quality control (QC) is a systematic approach to ensuring that products and services meet customer requirements. It is an essential part of manufacturing and industry, as it helps to improve product quality, customer satisfaction, and profitability. Quality practitioners generally apply control charts to monitor the industrial process, among many other statistical process control tools, and to detect changes. New developments in control charting schemes for high-quality monitoring are the need of the hour. In this paper, we have enhanced the performance of the mixed homogeneously weighted moving average (HWMA)-cumulative sum (CUSUM) control chart by using the auxiliary information-based (AIB) regression estimator and named it $MHC_{AIB}$. The proposed $MHC_{AIB}$ chart provided an unbiased and more efficient estimator of the process location. The various measures of the run length are used to judge the performance of the proposed $MHC_{AIB}$ and to compare it with existing AIB charts like $CUSUM_{AIB}$, $EWMA_{AIB}$, $MEC_{AIB}$ (mixed AIB EWMA-CUSUM), and $HWMA_{AIB}$. The Run length (RL) based performance comparisons indicate that the $MHC_{AIB}$ chart performs relatively better in monitoring small to moderate shifts over its competitor's charts. It is shown that the chart's performance improves with the increase in correlation between the study variable and the auxiliary variable. An illustrative application of the proposed $MHC_{AIB}$ chart is also provided to show its implementation in practical situations.

## 1. Introduction

Statistical process control and monitoring (SPCM) consists of several statistical tools, and control charts are considered the most efficient. The control charts resolve irregular deviations from the required standards in manufacturing and industrial processes. The memory-less and memory types are the two core divisions of the control charts (cf. Montgomery [1]). Shewhart [2] proposed memory-less control charts, which use only current sample information for process monitoring. The memory type charting procedures, for instance, the cumulative sum (CUSUM), the exponentially weighted moving average (EWMA), the progressive mean (PM), and the homogeneously weighted moving average (HWMA) were developed by Page [3], Roberts [4], Abbas et al. [5] and Abbas [6] respectively and the monitoring statistics of these charts grasp earlier sample information along with the recent information.

On control charts, various types of extensions have been introduced in the literature on the SPCM. Combining two control charts also improved the efficiency of the control charts. Lucas

**Data Availability Statement:** All relevant data are within the manuscript and its Supporting information files.

**Funding:** The author(s) received no specific funding for this work.

**Competing interests:** The authors have declared that no competing interests exist.

[7] and Lucas and Saccucci [8] suggested the combined design structure of the Shewhart-EWMA and Shewhart-CUSUM charts, respectively. Shamma and Shamma [9] proposed a double EWMA chart. Mixed design structures of EWMA-CUSUM (MEC) and CUSUM-EWMA (MCE) charts were suggested by Abbas et al. [10] and Zaman et al. [11] respectively. Motivated by the study of Shamma and Shamma [8], double PM and HWMA charts were suggested by Abbas et al. [12] and Abid et al. [13] respectively. A mixture of PM and EWMA charts was proposed by Abbas et al. [14]. Taking inspiration from Abbas et al. [9] and Abid et al. [15] developed a mixed HWMA-CUSUM (MHC) chart in which statistic of the CUSUM chart runs as the output and MHC chart outperforms against the EWMA, HWMA, and MEC charts.

In sample surveys, the precision of the estimates of the population parameters can be increased by using auxiliary information. The auxiliary variable is a variable known for all units of the population but not a variable under study. The auxiliary information-based (AIB) charts are usually developed using regression and ratio estimators to monitor the process variable effectively. In the SPCM literature, much work has been done related to the AIB charts. Riaz [16]and Riaz [17] proposed a regression estimator-based Shewhart $_{(AIB)}$ chart for monitoring process location and dispersion, respectively. The regression EWMA$_{AIB}$ chart was proposed by Abbas et al. [18] and the EWMA$_{AIB}$ performed well against the usual EWMA chart without the AIB information. Abbas [19] suggested the CUSUMAIB chart performed relatively better than the usual CUSUM chart. Ahmad et al. [20] suggested some AIB charts for the autocorrelated processes. Adegoke et al. [21] designed a regression HWMA$_{AIB}$ chart when the process variable is investigated under normal and non-normal environments and revealed that the HWMA$_{AIB}$ chart is more powerful than the EWMA$_{AIB}$ and CUSUM$_{AIB}$ charts. Sanusi et al. [22] suggested various ratio estimators based on EWMA charts. The regression PM$_{AIB}$ chart was suggested by Abbas et al. [12] under zero-state and steady-state processes. Interested readers can see the work of Ahmad et al. [20], Haq and Khoo [23], Abbasi and Haq [24], Noor-ul-Amin et al. [25], and Hussain et al. [26] on AIB charts. Dirbaz et al. [27] suggested two new AIB-based control charts, AIB-MEWMA and AIB-DMEWMA charts, to detect shifts in model parameters. Arslan et al. [28] designed a sensitive homogeneously weighted moving average chart using two supplementary variables (hereafter, TAHWMA), which is an efficient and unbiased estimator for the process mean if the two supplementary variables correlate with the study variable.

In the SPCM literature, very little work is available on AIB mixed memory control charts. Recently, Anwar et al. [29] designed a regression estimator based MEC$_{AIB}$ and MCE$_{AIB}$ charts for prompt detection of persistent changes, and the MEC$_{AIB}$ chart is more effective than the MCE$_{AIB}$ chart and as well as the EWMA$_{AIB}$ and CUSUM$_{AIB}$ charts. Adegoke et al. [21] revealed that the performance of the regression estimator is relatively better than the ratio estimator. The core focus of this study is to propose an efficient mixed memory chart under the scenario of the regression estimator. So, this study proposes a new regression estimator based MHC chart labeled as MHC$_{AIB}$ for detecting persistent deviations in the process location. The MHC$_{AIB}$ chart is a mixture of the HWMA$_{AIB}$ and the usual CUSUM chart. In the recent age of development, improvements in quality assurance techniques are the need of the hour. In context, we have developed a new chart showing visible improvement in detecting shifts. Even a minute change and deviation in the quality can be a big hurdle in many industrial processes like lifesaving drugs, substrate manufacturing, missile equipment, etc. In some industrial and manufacturing processes, the auxiliary information is also recorded along with the under-study variables for different tasks. This information can be used to improve the control chart design without imparting any additional financial burden to the entrepreneur. We can use this information for the improvement of design. Our proposed chart is shown to have improved results compared to its counterparts and can be used for high-quality monitoring in different industrial applications.

The rest of the article is outlined as follows: the next section offers the structure of the MHC and the $\text{MHC}_{\text{AIB}}$ charts, along with the RL evaluation of the proposed $\text{MHC}_{\text{AIB}}$ chart. The performance evaluation and RL comparisons of the $\text{MHC}_{\text{AIB}}$ chart against the competitor's charts are delivered in Section 3. A numerical example of the $\text{MHC}_{\text{AIB}}$ and existing charts are offered in Section 4, Section 5 gives the limitation of the study, and the article ends with a conclusion and recommendations.

## 2. The Mixed HWMA-CUSUM (MHC) and the proposed Mixed HWMA-CUSUM with auxiliary information ($\text{MHC}_{\text{AIB}}$) control charts

This section includes a description of the construction of the existing Mixed HWMA-CUSUM (MHC) and the proposed Mixed HWMA-CUSUM with additional information $\text{MHC}_{\text{AIB}}$ charts:

### 2.1. The MHC chart

Let $z_{ij}$ is the variable of interest follows the normal distribution, i.e., $z_{ij} \sim N\left(\mu_z, \ \sigma_z^2\right)$ where $\mu_z$ and $\sigma_z^2$ is the in-control (IC) mean and variance of the process variable, respectively, $i = 1, 2, 3, \ldots$ and $j = 1, 2, 3, \ldots, n$. Abbas [6] suggested the statistic of the HWMA chart as follows:

$$H_i = \theta \bar{z}_i + (1 - \theta)\bar{\bar{z}}_{i-1} \tag{1}$$

Where $\theta$ is the smoothing parameter ($\theta \in (0, 1]$), $\bar{\bar{z}}_0 = \mu_z$, and $\bar{\bar{z}}_{i-1} = \frac{\sum_{m=1}^{i-1} \bar{z}_m}{i-1}$. The IC mean and variance for $H_i$ are as follows (cf. Abbas [6]):

$$
\left.\begin{array}{l} Mean(H_i) = \mu_z \\ Var(H_i) = \dfrac{\theta^2 \sigma_z^2}{n} \ if \ i = 1 \end{array}\right) And \left.\begin{array}{l} Mean(H_i) = \mu_z \\ Var(H_i) = \dfrac{\theta^2 \sigma_z^2}{n} + (1-\theta)^2 \dfrac{\sigma_z^2}{n(t-1)} \ if \ i > 1 \end{array}\right)
$$

Abid et al. [15] proposed the MHC chart by placing the statistic given in (1) with the CUSUM statistic, and the plotting statistics of the MHC chart are given as:

$$
\begin{aligned}
MHC_i^+ &= \max\left[0, (H_i - \mu_z) - K + MHC_{i-1}^+\right] \\
MHC_i^- &= \max\left[0, -(H_i - \mu_z) - K + MHC_{i-1}^-\right]
\end{aligned} \tag{2}
$$

Where $H_i$ is given in (1) and $MHC_i^+ = MHC_i^- = 0$. The K and $H$ are defined as (cf. Abid et al. [15])

$$
K = k\sqrt{Var(H_i)} = \left.\begin{array}{ll} k\sqrt{\dfrac{\theta^2 \sigma_z^2}{n}} & if \ i = 1 \\[3mm] k\sqrt{\dfrac{\theta^2 \sigma_z^2}{n} + \left(1-\theta^2\right)^2 \dfrac{\sigma_z^2}{n(i-1)}}, & if \ i > 1 \end{array}\right] \tag{3}
$$

$$
H = h\sqrt{Var(H_i)} = \left.\begin{array}{ll} h\sqrt{\dfrac{\theta^2 \sigma_z^2}{n}} & if \ i = 1 \\[3mm] h\sqrt{\dfrac{\theta^2 \sigma_z^2}{n} + \left(1-\theta^2\right)^2 \dfrac{\sigma_z^2}{n(i-1)}}, & if \ i > 1 \end{array}\right] \tag{4}
$$

And these are the parameters of the MHC chart. The process is considered to be out-of-control (OOC) if any value of $MHC_i^+$ or $MHC_i^-$ go beyond $H$; otherwise, it is considered to be in control.

## 2.2. The proposed MHC$_{AIB}$ chart

In most situations, there exists a positive/negative association between the study/process variable ($z_i$) and the auxiliary variable($x_i$). Let us assume that $x_i$ is strongly correlated with $z_i$ and the strength of the correlation between $x_i$ and $z_i$ is represented by $\rho_{zx}$. The pair of observations $x_i$ and $z_i$ follows the bivariate normal distribution, i.e., $(z_i, x_i) \sim BVN(\mu_z + \delta\sigma_z, \mu_x, \sigma_z, \sigma_x, \rho_{zx})$, where $\delta$ is mathematically written as $\delta = \frac{\sqrt{n}}{\sigma_z}|\mu_z - \mu_1|$, where $\mu_1$ is the shifted mean. The regression estimator suggested by Cochran [30] is as follows:

$$R_i = \bar{z}_i + b_{zx}(\mu_x - \bar{x}_i) \tag{5}$$

where $b_{zx} = \rho_{zx}\frac{\sigma_z}{\sigma_x}$ is the regression coefficient, $\bar{z}_i$ and $\bar{x}_i$ is the sample mean of the $z_i$ and $x_i$, respectively, and $\mu_x$ is the population mean of the $x_i$. The mean and variance of $R_i$ are given below (cf. Appendix A in S1 File):

$$\mu_R = \mu_z \text{ and } \sigma_R^2 = \frac{\sigma_z^2}{n}\left(1 - \rho_{zx}^2\right), \tag{6}$$

The plotting statistic of the proposed MHC$_{AIB}$ using (2) is defined as:

$$\begin{aligned} RHC_i^+ &= \max\left[0, (RH_i - \mu_z) - K + RHC_{i-1}^+\right] \\ RHC_i^- &= \max\left[0, -(RH_i - \mu_z) - K + RHC_{i-1}^-\right] \end{aligned} \tag{7}$$

Where

$$RH_i = \theta R_i + (1 - \theta)\bar{R}_{i-1} \tag{8}$$

$\theta$ is already defined above, $\bar{R}_0 = \mu_y$, $\bar{R}_{i-1} = \frac{\sum_{m=1}^{i-1} R_m}{i-1}$ and $RHC_i^+ = RHC_i^- = 0$. The MHC$_{AIB}$ chart depends on two parameters, i.e., K and H and which are mathematically written as: (cf. Abid et al. [15]):

$$K = k\sqrt{Var(RH_i)} = \begin{cases} k\sqrt{\frac{\theta^2}{n}\sigma_z^2\left(1 - \rho_{zx}^2\right)} & if\ i = 1 \\ k\sqrt{\left(\frac{\theta^2}{n} + \frac{\left(1 - \theta^2\right)^2}{n(i-1)}\right)\sigma_z^2\left(1 - \rho_{zx}^2\right)} & if\ i > 1 \end{cases} \tag{9}$$

$$H = h\sqrt{Var(RH_i)} = \begin{cases} h\sqrt{\frac{\theta^2}{n}\sigma_z^2\left(1 - \rho_{zx}^2\right)} & if\ i = 1 \\ h\sqrt{\left(\frac{\theta^2}{n} + \frac{\left(1 - \theta^2\right)^2}{n(i-1)}\right)\sigma_z^2\left(1 - \rho_{zx}^2\right)} & if\ i > 1 \end{cases} \tag{10}$$

$RHC_i^+$ and $RHC_i^-$ are plotted against the value of $H$ given in (10). If $RHC_i^+ > H$ or $RHC_i^- > H$, the process is assumed to be out of control (OOC); otherwise, it is in control (IC).

## 3. Performance evaluation of the MHC$_{AIB}$ chart

The help of a well-known measure assesses the performance of the proposed MHCAIB and its competitor charts called the average run length (ARL). The Average Run length can be defined as "The number of sample points before a control chart gives alarm is called run length (RL) and an average value of RL distribution is called ARL." The $ARL_{IC}$ and $ARL_{OOC}$ designated the

in-control (IC) and out-of-control (OOC) ARL. A chart having a lesser value of $ARL_{OOC}$ for a particular shift is designated to be better over the competitor chart at a certain shift in the process parameter(s) for a fixed value of $ARL_{IC}$. We have also included some other performance measures associated with RL, like (the standard deviation RL (SDRL) and the median RL (MDRL) (cf. Abid et al. [15])) and these measures are calculated through the Monte Carlo simulations approach. The computational algorithms are developed in the R programming language, and the computational algorithms' flow chart is presented in Fig 1.

The $MHC_{AIB}$ chart has the design parameters $n$, $\theta$, $\rho_{zx}$, $h$ and $k$. the $ARL_{OOC}$ values of the proposed $MHC_{AIB}$ chart against numerous choices of $\theta$ = 0.1, 0.25, 0.5, 0.75 and $\rho_{zx}$ = 0.25, 0.5,

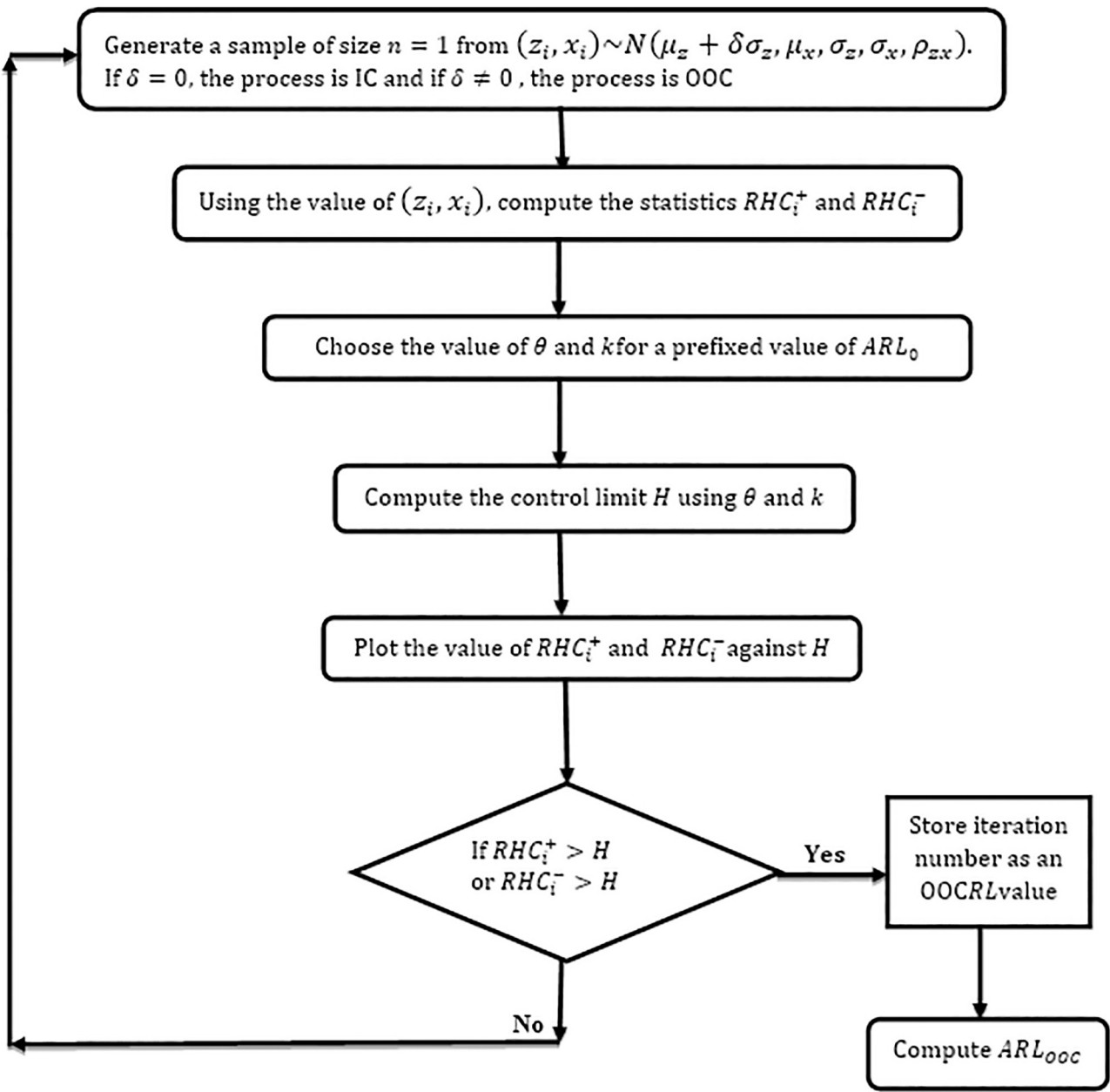

**Fig 1. The computational algorithm of the proposed MHC$_{AIB}$ chart.**

**Table 1. ARL profiles of the proposed $MHC_{AIB}$ chart for various choices of $\rho_{zx}$ when $\theta = 0.10$.**

| $\delta$ | $\rho_{zx}$ | | | | | | | | | | | |
|---|---|---|---|---|---|---|---|---|---|---|---|---|
| | 0.25 | | | 0.5 | | | 0.75 | | | 0.95 | | |
| | ARL | SDRL | MDRL | ARL | SDRL | MDRL | ARL | SDRL | MDRL | ARL | SDRL | MDRL |
| 0.05 | 117.43 | 330.51 | 26 | 107.81 | 283.32 | 26 | 81.51 | 174.14 | 26 | 39.74 | 51.08 | 21 |
| 0.075 | 79.94 | 170.53 | 26 | 71.00 | 141.90 | 25 | 55.88 | 91.46 | 24 | 26.49 | 26.75 | 17 |
| 0.1 | 60.77 | 103.54 | 24 | 53.52 | 83.82 | 24 | 41.03 | 55.58 | 21 | 19.98 | 16.85 | 14 |
| 0.125 | 48.51 | 71.08 | 22 | 43.52 | 59.43 | 22 | 33.67 | 38.87 | 19 | 16.01 | 11.66 | 12 |
| 0.15 | 41.10 | 52.68 | 22 | 37.92 | 49.11 | 21 | 27.52 | 27.53 | 18 | 13.36 | 8.68 | 11 |
| 0.175 | 35.35 | 42.97 | 20 | 31.33 | 37.19 | 19 | 23.90 | 22.44 | 16 | 11.64 | 6.85 | 10 |
| 0.2 | 30.63 | 33.86 | 19 | 26.87 | 28.15 | 17 | 21.38 | 19.15 | 15 | 10.19 | 5.45 | 9 |
| 0.25 | 24.90 | 24.31 | 16 | 22.18 | 19.80 | 15 | 17.10 | 13.03 | 13 | 8.42 | 3.70 | 7 |
| 0.5 | 12.44 | 7.64 | 10 | 11.20 | 6.37 | 9 | 8.82 | 4.01 | 8 | 5.06 | 1.20 | 5 |
| 0.75 | 8.73 | 4.00 | 8 | 7.91 | 3.28 | 7 | 6.42 | 2.11 | 6 | 4.00 | 0.69 | 4 |
| 1 | 6.93 | 2.48 | 6 | 6.37 | 2.05 | 6 | 5.26 | 1.34 | 5 | 3.41 | 0.52 | 3 |
| 1.5 | 5.17 | 1.30 | 5 | 4.83 | 1.07 | 5 | 4.13 | 0.73 | 4 | 3.01 | 0.09 | 3 |
| 2 | 4.36 | 0.84 | 4 | 4.09 | 0.72 | 4 | 3.53 | 0.56 | 4 | 2.88 | 0.34 | 3 |
| $h$ | 8.575 | | | | | | | | | | | |

0.75, 0.95 when the value of $k = 0.5$ are given in Tables 1–4. The proposed $MHC_{AIB}$ chart shows better performance against smaller values of $\theta$(for instance $\delta = 0.1$, $\rho_{zx} = 0.25$ when $\theta = 0.1$, $ARL_{OOC} = 103.54$ against $\theta = 0.75$, $ARL_{OOC} = 300.25$ (cf. Table 1 vs. Table 4)). An increase in $\rho_{zx}$ enhanced the efficiency of the $MHC_{AIB}$ chart (for instance $\delta = 0.05$, $\theta = 0.1$ when $\rho_{zx} = 0.25$, $ARL_{OOC} = 117.43$ against $\rho_{zx} = 0.95$, $ARL_{OOC} = 39.74$ (cf. Table 1)). There is a decrease in the OOC SDRL and MDRL values with the increase in $\rho_{zx}$ (for instance $\delta = 0.125$, $\theta = 0.1$ when $\rho_{zx} = 0.25$, $OOC\ SDRL = 71.08$, OOC MDRL = 22 against $\rho_{zx} = 0.95$, $OOC\ SDRL = 11.66$, OOC MDRL = 12 (cf. Table 1)).

**Table 2. ARL profiles of the proposed $MHC_{AIB}$ chart for various choices of $\rho_{zx}$ when $\theta = 0.25$.**

| $\delta$ | $\rho_{zx}$ | | | | | | | | | | | |
|---|---|---|---|---|---|---|---|---|---|---|---|---|
| | 0.25 | | | 0.5 | | | 0.75 | | | 0.95 | | |
| | ARL | SDRL | MDRL | ARL | SDRL | MDRL | ARL | SDRL | MDRL | ARL | SDRL | MDRL |
| 0.05 | 262.14 | 578.16 | 35 | 229.30 | 490.45 | 36 | 169.30 | 341.59 | 33 | 56.88 | 86.91 | 23 |
| 0.075 | 162.23 | 321.10 | 33 | 140.22 | 272.26 | 31 | 93.42 | 166.15 | 28 | 32.15 | 38.09 | 18 |
| 0.1 | 110.16 | 201.02 | 28 | 93.41 | 160.70 | 28 | 63.36 | 101.08 | 24 | 21.60 | 21.46 | 14 |
| 0.125 | 77.61 | 132.10 | 26 | 67.04 | 106.21 | 25 | 44.52 | 60.35 | 21 | 16.15 | 13.88 | 11 |
| 0.15 | 60.21 | 92.09 | 24 | 51.56 | 74.87 | 22 | 33.92 | 42.41 | 18 | 12.87 | 9.66 | 10 |
| 0.175 | 48.25 | 69.14 | 21 | 40.13 | 52.29 | 20 | 27.69 | 31.96 | 16 | 10.84 | 7.19 | 9 |
| 0.2 | 40.05 | 54.03 | 20 | 33.76 | 41.19 | 18 | 23.35 | 24.30 | 15 | 9.33 | 5.57 | 8 |
| 0.25 | 28.71 | 33.30 | 17 | 24.84 | 26.76 | 15 | 17.40 | 15.68 | 12 | 7.56 | 3.79 | 7 |
| 0.5 | 12.06 | 8.82 | 9 | 10.46 | 6.82 | 8 | 7.99 | 4.30 | 7 | 4.30 | 1.12 | 4 |
| 0.75 | 7.78 | 4.05 | 7 | 6.97 | 3.22 | 6 | 5.54 | 2.01 | 5 | 3.36 | 0.57 | 3 |
| 1 | 6.00 | 2.42 | 5 | 5.46 | 1.99 | 5 | 4.45 | 1.23 | 4 | 3.01 | 0.32 | 3 |
| 1.5 | 4.40 | 1.21 | 4 | 4.06 | 0.99 | 4 | 3.46 | 0.63 | 3 | 2.40 | 0.51 | 2 |
| 2 | 3.64 | 0.75 | 4 | 3.43 | 0.61 | 3 | 3.06 | 0.33 | 3 | 1.78 | 0.44 | 2 |
| $h$ | 6.625 | | | | | | | | | | | |

**Table 3. ARL profiles of the proposed $MHC_{AIB}$ chart for various choices of $\rho_{zx}$ when $\theta = 0.5$.**

| $\delta$ | $\rho_{zx}$ | | | | | | | | | | | |
|---|---|---|---|---|---|---|---|---|---|---|---|---|
| | 0.25 | | | 0.5 | | | 0.75 | | | 0.95 | | |
| | ARL | SDRL | MDRL | ARL | SDRL | MDRL | ARL | SDRL | MDRL | ARL | SDRL | MDRL |
| 0.05 | 370.80 | 521.37 | 161 | 359.00 | 493.57 | 158 | 292.51 | 408.95 | 130 | 116.32 | 151.77 | 57 |
| 0.075 | 290.12 | 402.43 | 128 | 261.84 | 351.97 | 124 | 190.26 | 249.90 | 91 | 57.40 | 68.82 | 32 |
| 0.1 | 221.19 | 298.70 | 103 | 187.51 | 248.80 | 89 | 126.50 | 161.65 | 64 | 33.70 | 35.31 | 21 |
| 0.125 | 156.20 | 205.41 | 75 | 135.08 | 173.55 | 68 | 87.53 | 108.64 | 46 | 22.93 | 21.24 | 16 |
| 0.15 | 123.84 | 160.76 | 62 | 100.98 | 125.41 | 53 | 62.99 | 75.14 | 35 | 16.97 | 14.19 | 12 |
| 0.175 | 94.60 | 119.63 | 48 | 77.40 | 93.43 | 42 | 48.38 | 54.90 | 28 | 13.33 | 10.19 | 10 |
| 0.2 | 73.09 | 86.36 | 40 | 61.04 | 72.44 | 34 | 36.64 | 39.07 | 22 | 10.94 | 7.57 | 9 |
| 0.25 | 50.02 | 57.60 | 29 | 41.24 | 44.91 | 25 | 25.54 | 24.71 | 17 | 8.16 | 4.74 | 7 |
| 0.5 | 14.98 | 12.03 | 11 | 12.90 | 9.79 | 10 | 8.74 | 5.25 | 7 | 3.98 | 1.22 | 4 |
| 0.75 | 8.46 | 4.97 | 7 | 7.48 | 4.09 | 6 | 5.51 | 2.34 | 5 | 2.95 | 0.63 | 3 |
| 1 | 6.04 | 2.83 | 5 | 5.40 | 2.26 | 5 | 4.15 | 1.32 | 4 | 2.40 | 0.51 | 2 |
| 1.5 | 4.09 | 1.31 | 4 | 3.72 | 1.05 | 3 | 3.06 | 0.67 | 3 | 1.90 | 0.33 | 2 |
| 2 | 3.26 | 0.76 | 3 | 3.02 | 0.65 | 3 | 2.51 | 0.53 | 2 | 1.36 | 0.48 | 1 |
| $h$ | 5.536 | | | | | | | | | | | |

The performance assessment of the MHC$_{AIB}$ chart in the form of a line graph is given in Fig 2A–2D against various choices of $\theta$ and $\rho_{zx}$. A decrease in $\theta$ enhanced the efficacy of the MHC$_{AIB}$ chart and vice versa (cf. Fig 2A vs. Fig 2D). Also, an increase in the value of $\rho_{zx}$ improved the sensitivity of the MHC$_{AIB}$ chart and vice versa. It means that a higher correlation coefficient value increases the suggested chart's efficiency. (cf. Fig 2A–2D).

## 3.1. Comparisons

This section offered the OOC performance assessment of the proposed MHC$_{AIB}$ with the CUSUM$_{AIB}$, EWMA$_{AIB}$, HWMA$_{AIB}$, and MEC$_{AIB}$ suggested by Abbas et al. [18], Abbas [19],

**Table 4. ARL profiles of the proposed $MHC_{AIB}$ chart for various choices of $\rho_{zx}$ when $\theta = 0.75$.**

| $\delta$ | $\rho_{zx}$ | | | | | | | | | | | |
|---|---|---|---|---|---|---|---|---|---|---|---|---|
| | 0.25 | | | 0.5 | | | 0.75 | | | 0.95 | | |
| | ARL | SDRL | MDRL | ARL | SDRL | MDRL | ARL | SDRL | MDRL | ARL | SDRL | MDRL |
| 0.05 | 430.62 | 457.84 | 284.5 | 415.16 | 442.92 | 276 | 371.10 | 407.80 | 240 | 190.93 | 200.22 | 126 |
| 0.075 | 372.53 | 405.59 | 239 | 350.00 | 371.77 | 232.5 | 279.72 | 304.95 | 180 | 100.69 | 103.63 | 67 |
| 0.1 | 300.25 | 326.22 | 191 | 272.43 | 292.73 | 179 | 203.04 | 217.61 | 131 | 58.76 | 56.94 | 40 |
| 0.125 | 242.70 | 262.71 | 158 | 213.69 | 230.35 | 137.5 | 149.28 | 159.03 | 97 | 37.43 | 34.60 | 26 |
| 0.15 | 199.04 | 208.87 | 128 | 169.69 | 181.26 | 110 | 110.07 | 118.21 | 71 | 26.25 | 22.43 | 19 |
| 0.175 | 162.29 | 171.48 | 105.5 | 132.78 | 139.83 | 88 | 82.65 | 83.97 | 56 | 19.62 | 15.02 | 15 |
| 0.2 | 128.11 | 137.00 | 83 | 106.64 | 110.47 | 70 | 65.93 | 66.42 | 44 | 15.48 | 11.20 | 12 |
| 0.25 | 88.92 | 90.87 | 59 | 71.37 | 71.88 | 47 | 41.77 | 39.33 | 29 | 10.73 | 6.46 | 9 |
| 0.5 | 22.51 | 18.08 | 17 | 18.40 | 14.13 | 14 | 11.84 | 7.59 | 10 | 4.36 | 1.53 | 4 |
| 0.75 | 11.29 | 6.94 | 9 | 9.54 | 5.37 | 8 | 6.61 | 3.09 | 6 | 2.92 | 0.77 | 3 |
| 1 | 7.47 | 3.64 | 7 | 6.47 | 2.99 | 6 | 4.65 | 1.72 | 4 | 2.29 | 0.49 | 2 |
| 1.5 | 4.54 | 1.66 | 4 | 4.01 | 1.33 | 4 | 3.06 | 0.83 | 3 | 1.82 | 0.39 | 2 |
| 2 | 3.36 | 0.98 | 3 | 3.05 | 0.82 | 3 | 2.40 | 0.55 | 2 | 1.24 | 0.43 | 1 |
| $h$ | 5.187 | | | | | | | | | | | |

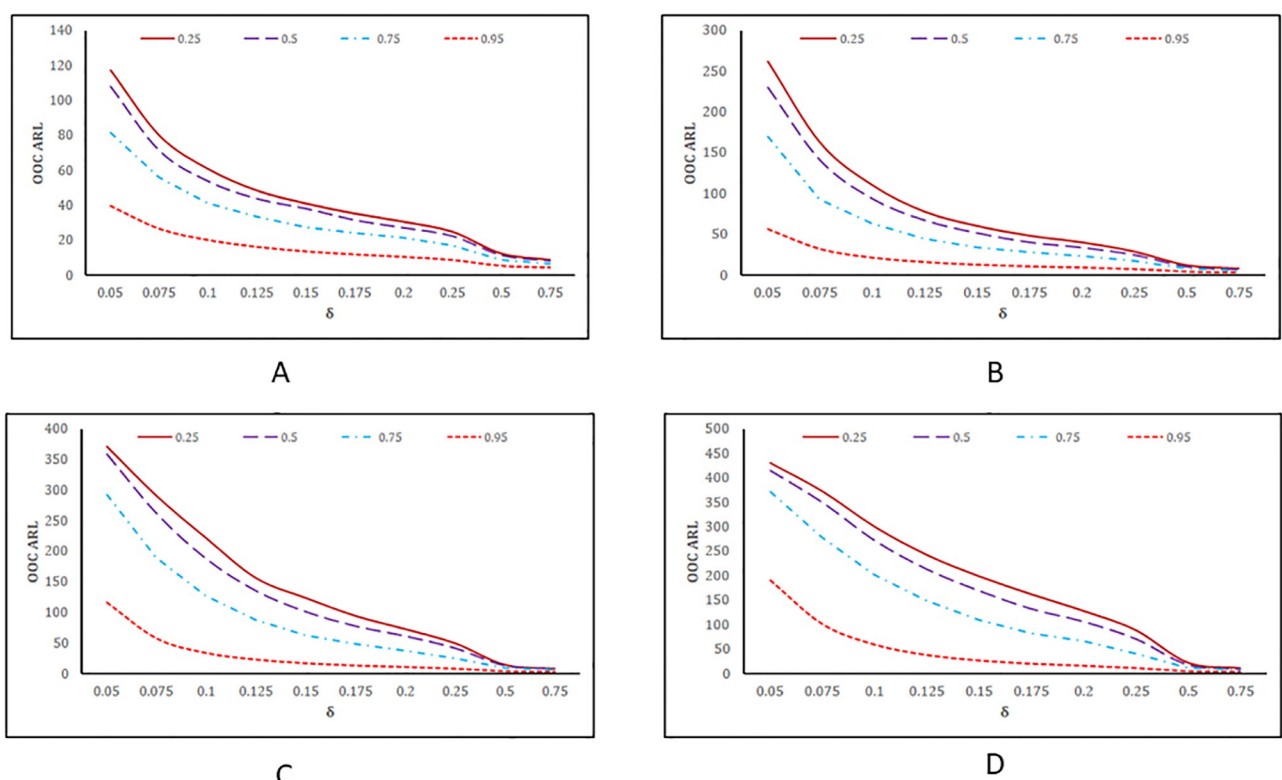

**Fig 2.** $ARL_{OOC}$ **values based graphical comparison of the proposed MHC$_{AIB}$ chart for various choices of $\rho_{zx}$.** (A). when $\theta = 0.1$, (B). when $\theta = 0.25$, (C). when $\theta = 0.5$ and (D) when $\theta = 0.75$.

Sanusi et al. [22], and Anwar et al. [27] respectively. Moreover, we have also expressed these comparisons as a percentage decrease in ARL. ($ARL_{PD}$) and mathematically $ARL_{PD}$ is defined as $\left( \frac{ARL_{IC} - ARL_{OOC}}{ARL_{IC}} \right) \times 100\%$. The chart with the highest $ARL_{PD}$ value is labeled an efficient chart for that specific shift.

**3.1.1. MHC$_{AIB}$ versus CUSUM$_{AIB}$.** Abbas et al. [18] suggested the CUSUM$_{AIB}$ chart and the $ARL_{OOC}$ results of the CUSUM$_{AIB}$ chart are provided in Tables 5 and 6 against various choices of $\theta$, $\rho_{zx}$ and $\delta$. The proposed MHC$_{AIB}$ chart compromises enhanced performance over the CUSUM$_{AIB}$ chart for all selections of $\theta$, $\rho_{zx}$, and $\delta$ (for instance when $\delta = (0.05, 0.1, 0.2)$, $\theta = 0.1$ and $\rho_{zx} = 0.5$, the MHC$_{AIB}$ $ARL_{OOC} = (108, 54, 27)$ and the CUSUM$_{AIB}$ $ARL_{OOC} = (374, 220, 92)$ (cf. Table 5). Also, at $\delta = 0.05$, the $ARL_{PD}$ in CUSUM$_{AIB}$ and MHC$_{AIB}$ charts are 25.2%and 78.4%, when $\theta = 0.1$ and $\rho_{zx} = 0.5$, respectively.

**3.1.2. MHC$_{AIB}$ versus EWMA$_{AIB}$.** Abbas [19] recommended the EWMA$_{AIB}$ chart and the results of $ARL_{OOC}$ values of the EWMA$_{AIB}$ chart are specified in Tables 5 and 6. The suggested MHC$_{AIB}$ chart displays comparatively better performance over the EWMA$_{AIB}$ chart when $\delta \leq 0.75$ (for instance, at $\delta = (0.05, 0.1, 0.2)$, $\theta = 0.1$, and $\rho_{zx} = 0.5$, the MHC$_{AIB}$ $ARL_{OOC} = (108, 54, 27)$ and the EWMA$_{AIB}$ $ARL_{OOC} = (421, 282, 118)$ (cf. Table 5) and when $\theta = 0.25$, the MHC$_{AIB}$ $ARL_{OOC} = (229, 93, 34)$ and the EWMA$_{AIB}$ $ARL_{OOC} = (455, 361, 190)$ (cf. Table 6). Furthermore, at $\delta = 0.1$, the $ARL_{PD}$ in AIB-EWMA and AIB-MHC charts are 43.6% and 89.2% when $\theta = 0.1$ and $\rho_{zx} = 0.5$, respectively.

**3.1.3. MHC$_{AIB}$ versus HWMA$_{AIB}$.** Sanusi et al. [22] introduced the HWMA$_{AIB}$ chart and the results of $ARL_{OOC}$ values of HWMA$_{AIB}$ chart are provided in Tables 5 and 6 for several

**Table 5. ARL comparisons between proposed MHC$_{AIB}$ and existing charts for various choices of $\rho_{zx}$ when $\theta = 0.1$.**

| $\delta$ | CUSUM$_{AIB}$ | | | EWMA$_{AIB}$ | | | HWMA$_{AIB}$ | | | MEC$_{AIB}$ | | | MHC$_{AIB}$ | | |
|---|---|---|---|---|---|---|---|---|---|---|---|---|---|---|---|
| | $\rho_{zx}$ | | | $\rho_{zx}$ | | | $\rho_{zx}$ | | | $\rho_{zx}$ | | | $\rho_{zx}$ | | |
| | 0.5 | 0.75 | 0.95 | 0.5 | 0.75 | 0.95 | 0.5 | 0.75 | 0.95 | 0.5 | 0.75 | 0.95 | 0.5 | 0.75 | 0.95 |
| 0.0 | 498 | 502 | 501 | 501 | 504 | 499 | 502 | 502 | 502 | 499 | 502 | 500 | 500 | 501 | 500 |
| 0.05 | 374 | 320 | 150 | 421 | 378 | 200 | 374 | 315 | 149 | 370 | 303 | 143 | 108 | 82 | 40 |
| 0.075 | 287 | 223 | 88 | 350 | 288 | 111 | 286 | 222 | 87 | 275 | 213 | 84 | 71 | 56 | 26 |
| 0.1 | 220 | 161 | 60 | 282 | 214 | 67 | 216 | 160 | 57 | 212 | 153 | 58 | 54 | 41 | 20 |
| 0.125 | 170 | 122 | 45 | 228 | 160 | 44 | 168 | 120 | 40 | 162 | 114 | 45 | 44 | 34 | 16 |
| 0.15 | 135 | 95 | 36 | 182 | 121 | 31 | 135 | 94 | 31 | 128 | 91 | 37 | 38 | 28 | 13 |
| 0.175 | 111 | 78 | 30 | 145 | 94 | 23 | 110 | 75 | 24 | 106 | 75 | 31 | 31 | 24 | 12 |
| 0.2 | 92 | 65 | 26 | 118 | 74 | 18 | 91 | 62 | 19 | 88 | 62 | 28 | 27 | 21 | 10 |
| 0.25 | 69 | 49 | 20 | 81 | 49 | 12 | 66 | 44 | 13 | 66 | 48 | 23 | 22 | 17 | 8 |
| 0.5 | 29 | 22 | 10 | 22 | 13 | 4 | 23 | 15 | 5 | 31 | 24 | 13 | 11 | 9 | 5 |
| 0.75 | 18 | 14 | 6 | 11 | 7 | 2 | 12 | 8 | 3 | 21 | 17 | 10 | 8 | 6 | 4 |
| 1 | 14 | 10 | 5 | 6 | 4 | 1 | 7 | 5 | 2 | 17 | 14 | 8 | 6 | 5 | 3 |
| 1.5 | 9 | 7 | 3 | 3 | 2 | 1 | 4 | 3 | 1 | 12 | 10 | 6 | 5 | 4 | 3 |
| 2 | 7 | 5 | 3 | 2 | 2 | 1 | 3 | 2 | 1 | 10 | 8 | 5 | 4 | 4 | 3 |
| | $k = 0.5, h = 5.071$ | | | $C = 2.824$ | | | $C = 2.936$ | | | $k = 0.5, h = 37.35$ | | | $k = 0.5, h = 8.575$ | | |

choices of $\theta$, $\rho_{zx}$, and $\delta$. The suggested MHC$_{AIB}$ chart shows a better $ARL_{OOC}$ performance against the HWMA$_{AIB}$ chart for all choices of $\theta$ and $\rho_{zx}$ when $\delta \leq 0.75$ (for instance at $\delta = (0.05, 0.1, 0.2)$, $\theta = 0.1$, and $\rho_{zx} = 0.75$, the MHC$_{AIB}$ $ARL_{OOC} = (82, 41, 21)$ and the HWMA$_{AIB}$ $ARL_{OOC} = (315, 160, 62)$ (cf. Table 5) and when $\theta = 0.25$, the MHC$_{AIB}$ $ARL_{OOC} = (169, 63, 23)$ and the HWMA$_{AIB}$ $ARL_{OOC} = (385, 224, 83)$, (cf. Table 6). Moreover, at $\delta = 0.1$, the $ARL_{PD}$ in HWMA$_{AIB}$ and MHC$_{AIB}$ charts are 55.2% and 87.4% when $\theta = 0.25$ and $\rho_{zx} = 0.75$, respectively.

**Table 6. ARL comparisons between proposed MHC$_{AIB}$ and existing charts for various choices of $\rho_{zx}$ when $\theta = 0.25$.**

| $\delta$ | CUSUM$_{AIB}$ | | | EWMA$_{AIB}$ | | | HWMA$_{AIB}$ | | | MEC$_{AIB}$ | | | MHC$_{AIB}$ | | |
|---|---|---|---|---|---|---|---|---|---|---|---|---|---|---|---|
| | $\rho_{zx}$ | | | $\rho_{zx}$ | | | $\rho_{zx}$ | | | $\rho_{zx}$ | | | $\rho_{zx}$ | | |
| | 0.5 | 0.75 | 0.95 | 0.5 | 0.75 | 0.95 | 0.5 | 0.75 | 0.95 | 0.5 | 0.75 | 0.95 | 0.5 | 0.75 | 0.95 |
| 0.0 | 498 | 502 | 501 | 499 | 500 | 500 | 501 | 500 | 501 | 502 | 499 | 500 | 499 | 500 | 499 |
| 0.05 | 374 | 320 | 150 | 455 | 429 | 285 | 425 | 385 | 209 | 396 | 337 | 161 | 229 | 169 | 57 |
| 0.075 | 287 | 223 | 88 | 414 | 369 | 179 | 355 | 296 | 120 | 307 | 241 | 89 | 140 | 93 | 32 |
| 0.1 | 220 | 161 | 60 | 361 | 299 | 114 | 290 | 224 | 75 | 238 | 173 | 58 | 93 | 63 | 22 |
| 0.125 | 170 | 122 | 45 | 311 | 242 | 75 | 235 | 171 | 51 | 187 | 128 | 42 | 67 | 45 | 16 |
| 0.15 | 135 | 95 | 36 | 265 | 194 | 52 | 191 | 132 | 36 | 145 | 97 | 33 | 52 | 34 | 13 |
| 0.175 | 111 | 78 | 30 | 225 | 157 | 37 | 154 | 104 | 27 | 116 | 77 | 27 | 40 | 28 | 11 |
| 0.2 | 92 | 65 | 26 | 190 | 127 | 27 | 128 | 83 | 22 | 96 | 64 | 23 | 34 | 23 | 9 |
| 0.25 | 69 | 49 | 20 | 137 | 84 | 17 | 90 | 56 | 14 | 68 | 46 | 18 | 25 | 17 | 8 |
| 0.5 | 29 | 22 | 10 | 35 | 19 | 4 | 26 | 16 | 4 | 26 | 19 | 9 | 10 | 8 | 4 |
| 0.75 | 18 | 14 | 6 | 14 | 8 | 2 | 13 | 8 | 2 | 16 | 12 | 7 | 7 | 6 | 3 |
| 1 | 14 | 10 | 5 | 8 | 5 | 1 | 8 | 5 | 2 | 12 | 10 | 5 | 5 | 4 | 3 |
| 1.5 | 9 | 7 | 3 | 4 | 2 | 1 | 4 | 3 | 1 | 9 | 7 | 4 | 4 | 3 | 2 |
| 2 | 7 | 5 | 3 | 2 | 2 | 1 | 3 | 2 | 1 | 7 | 6 | 3 | 3 | 3 | 2 |
| | $k = 0.5, h = 5.071$ | | | $C = 3.001$ | | | $C = 3.075$ | | | $k = 0.5, h = 20.17$ | | | $k = 0.5, h = 6.625$ | | |

**3.1.4. MHC$_{AIB}$ versus MEC$_{AIB}$.** Anwar et al. [29] suggested the MEC$_{AIB}$ chart and the results of $ARL_{OOC}$ values of MEC$_{AIB}$ chart are specified in Tables 5 and 6. The suggested MHC$_{AIB}$ chart proposes reasonably superior performance against the MEC$_{AIB}$ chart for all choices of $\theta$, $\rho_{zx}$ and $\delta$ (for instance at $\delta = 0.05, 0.1, 0.2$, $\theta = 0.1$ and $\rho_{zx} = 0.75$, the MHC$_{AIB}$ $ARL_{OOC} = (82, 41, 21)$ and the MEC$_{AIB}$ $ARL_{OOC} = (303, 153, 62)$ (cf. Table 5) and when $\theta = 0.25$, the MHC$_{AIB}$ $ARL_{OOC} = (169, 63, 23)$ and the MEC$_{AIB}$ $ARL_{OOC} = (337, 173, 64)$ (cf. Table 6). Also, at $\delta = 0.1$, the $ARL_{PD}$ in AIB-MEC and AIB-MHC charts are 65.4% and 87.4% when $\theta = 0.25$ and $\rho_{zx} = 0.75$, respectively.

## 3.2. Graphical comparisons based on $ARL_{OOC}$

The $ARL_{OOC}$ based graphical comparisons of the proposed MHC$_{AIB}$, EWMA$_{AIB}$, HWMA$_{AIB}$, and MEC$_{AIB}$ charts are presented in Fig 3A–3C for various choices $\rho_{zx}$. The proposed MHC$_{AIB}$ chart offers inferior performance for the selected choices of $\theta$ and $\rho_{zx}$ (cf. Fig 3A–3C). Additionally, the proposed MHC$_{AIB}$ chart displays superiority over the EWMA$_{AIB}$, HWMA$_{AIB}$, and MEC$_{AIB}$ charts as the value of $\rho_{zx}$ is increased (cf. Fig 3A vs. Fig 3D).

# 4. An illustrative example

Based on the simulated dataset, this section offers an illustrative example of the proposed MHCAIB, HWMAAIB, and MECAIB charts. This dataset consists of 20 pairs of observations which are obtained from the bivariate normal distribution, i.e., $(z_i, x_i) \sim N(\mu_z + \delta\sigma_z, \mu_x, \sigma_z, \sigma_x, \rho_{zx})$ by using the following values of $\mu_z = 0$, $\delta = 0.50$, $\mu_x = 0$, $\sigma_z = 1$, $\sigma_x = 1$ and $\rho_{zx} = 0.50$

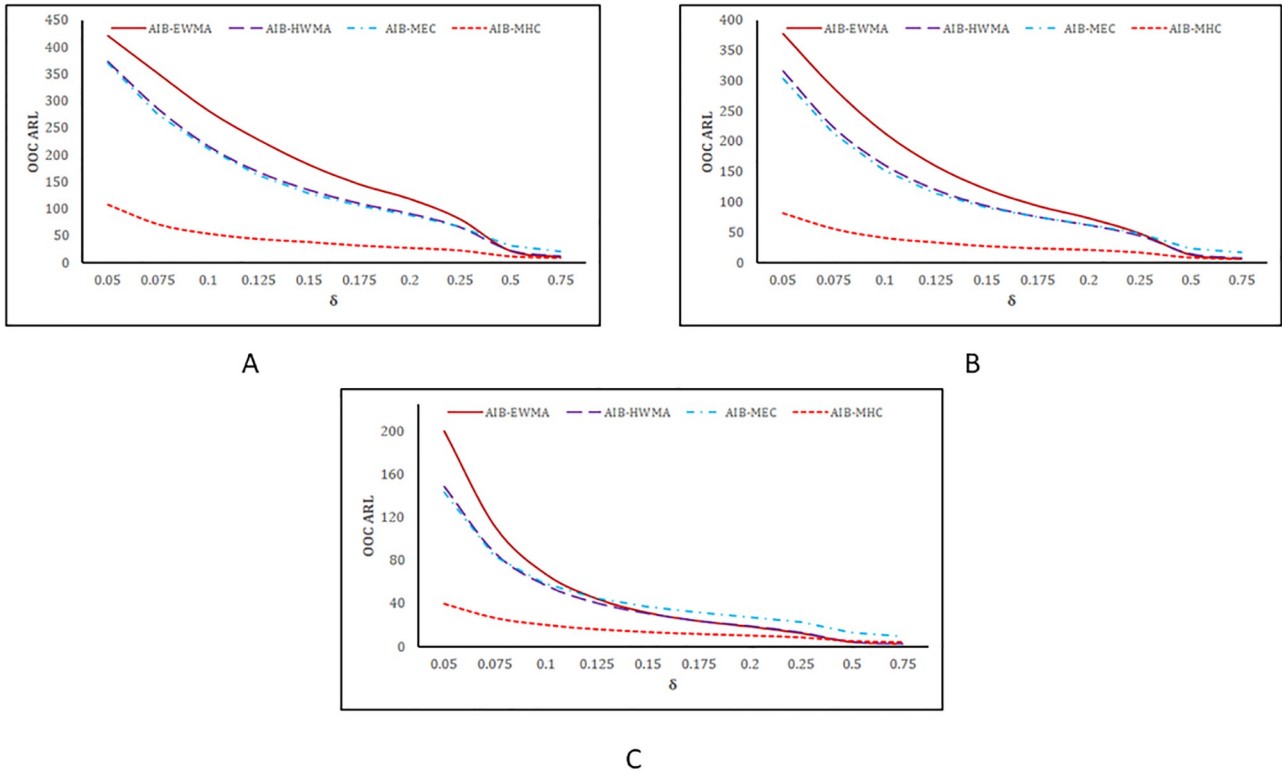

**Fig 3. $ARL_{OOC}$ values based graphical comparison between the proposed MHC$_{AIB}$ and existing charts for $\theta = 0.1$.** (A) when $\rho_{zx} = 0.5$, (B) when $\rho_{zx} = 0.75$, (C) when when $\rho_{zx} = 0.95$.

**Table 7. The plotting-statistic and the control limits values of the proposed MHC$_{AIB}$, MEC$_{AIB}$, and HWMA$_{AIB}$ charts.**

| $i$ | $z_i$ | $x_i$ | HWMA$_{AIB}$ chart | | | MEC$_{AIB}$ chart | | | MHC$_{AIB}$ chart | | |
|---|---|---|---|---|---|---|---|---|---|---|---|
| | | | HWMA$_{AIB}$ | UCL | LCL | $MEC^+$ | $MEC^-$ | $H$ | $RHC^+$ | $RHC^-$ | $H$ |
| 1 | 0.39 | -0.865 | 0.082 | 0.254 | -0.254 | 0.039 | 0 | 3.235 | 0.039 | 0 | 0.743 |
| 2 | -0.242 | -1.686 | 0.800 | 2.304 | -2.304 | 0.115 | 0 | 4.352 | 0.447 | 0 | 6.725 |
| 3 | -0.919 | -1.046 | 0.601 | 1.639 | -1.639 | 0.128 | 0 | 5.080 | 0.769 | 0 | 4.784 |
| 4 | -1.22 | -1.366 | 0.255 | 1.346 | -1.346 | 0.072 | 0 | 5.600 | 0.795 | 0 | 3.930 |
| 5 | 2.01 | 0.574 | 0.283 | 1.173 | -1.173 | 0.182 | 0 | 5.989 | 0.878 | 0 | 3.423 |
| 6 | 1.395 | 1.61 | 0.457 | 1.055 | -1.055 | 0.327 | 0 | 6.286 | 1.156 | 0 | 3.080 |
| 7 | 1.66 | 1.542 | 0.509 | 0.969 | -0.969 | 0.536 | 0 | 6.517 | 1.500 | 0 | 2.828 |
| 8 | -0.514 | 0.816 | 0.383 | 0.902 | -0.902 | 0.620 | 0 | 6.698 | 1.729 | 0 | 2.633 |
| 9 | -0.213 | -0.907 | 0.336 | 0.849 | -0.849 | 0.709 | 0 | 6.841 | 1.920 | 0 | 2.477 |
| 10 | -0.588 | -1.923 | 0.338 | 0.805 | -0.805 | 0.816 | 0 | 6.955 | 2.122 | 0 | 2.348 |
| 11 | 0.074 | 0.132 | 0.305 | 0.768 | -0.768 | 0.902 | 0 | 7.046 | 2.297 | 0 | 2.240 |
| 12 | 1.673 | 1.64 | 0.363 | 0.736 | -0.736 | 1.055 | 0 | 7.119 | 2.534 | 0 | 2.148 |
| 13 | 1.765 | 0.575 | 0.466 | 0.708 | -0.708 | 1.329 | 0 | 7.177 | 2.880 | 0 | 2.067 |
| 14 | 0.061 | -0.008 | 0.403 | 0.684 | -0.684 | 1.573 | 0 | 7.224 | 3.166 | 0 | 1.997 |
| 15 | 1.537 | -1.084 | 0.580 | 0.663 | -0.663 | 1.990 | 0 | 7.262 | 3.633 | 0 | 1.934 |
| 16 | -0.519 | -0.52 | 0.446 | 0.644 | -0.644 | 2.329 | 0 | 7.292 | 3.970 | 0 | 1.879 |
| 17 | 1.198 | -0.246 | 0.560 | 0.626 | -0.626 | 2.756 | 0 | 7.317 | 4.423 | 0 | 1.828 |
| 18 | 1.853 | 0.028 | 0.657 | 0.611 | -0.611 | 3.315 | 0 | 7.337 | 4.976 | 0 | 1.783 |
| 19 | 0.733 | 1.715 | 0.526 | 0.597 | -0.597 | 3.795 | 0 | 7.353 | 5.400 | 0 | 1.742 |
| 20 | 0.108 | 0.6 | 0.545 | 0.584 | -0.584 | 4.198 | 0 | 7.366 | 5.786 | 0 | 1.704 |

(cf. Abbas et al. [18]). This dataset is used to evaluate the shift-detecting capability of the proposed MHC$_{AIB}$, HWMA$_{AIB}$, and MEC$_{AIB}$ charts. The selected parameters for the practical implementation of the proposed MHC$_{AIB}$, HWMA$_{AIB}$, and MEC$_{AIB}$ charts are as follows: for the proposed MHC$_{AIB}$ chart $\theta = 0.1$, $k = 0.5$, and $h = 8.575$; for the MEC$_{AIB}$ chart $\theta = 0.1$, $k = 0.5$, and $h = 37.35$; for HWMA$_{AIB}$ chart $\theta = 0.1$, and $C = 2.936$ when $ARL_{IC} \approx 500$. The control limits and the plotting statistics of the MHC$_{AIB}$, HWMA$_{AIB}$, and MEC$_{AIB}$ charts are given in Table 7.

The HWMA$_{AIB}$ chart signals only one OOC point at the 18[th] sample (cf. Fig 4). The MEC$_{AIB}$ chart cannot produce any OOC signal (cf. Fig 5). Moreover, the proposed MHC$_{AIB}$ chart produces nine OOC signals from sample numbers 12 to 20 (cf. Fig 6), and this is a piece of evidence about the enhanced shift-detecting capability of the proposed MHC$_{AIB}$ chart against the HWMA$_{AIB}$, and MEC$_{AIB}$ charts.

## 5. Limitation

The proposed chart uses auxiliary information in its design, so it should be used only if there is a high correlation between the auxiliary variable and the study variable.

## 6. Conclusion and recommendations

A control chart is the most famous statistical process control and monitoring tool to detect irregular variations from ongoing processes. In this article, we have suggested a new regression estimator-based MHC chart labeled MHC$_{AIB}$ for monitoring persistent deviations in the process location. The $ARL_{OOC}$ performance of the suggested MHC$_{AIB}$ chart is compared with the CUSUM$_{AIB}$, EWMA$_{AIB}$, HWMA$_{AIB}$, and MEC$_{AIB}$, and the suggested MHC$_{AIB}$ chart performs

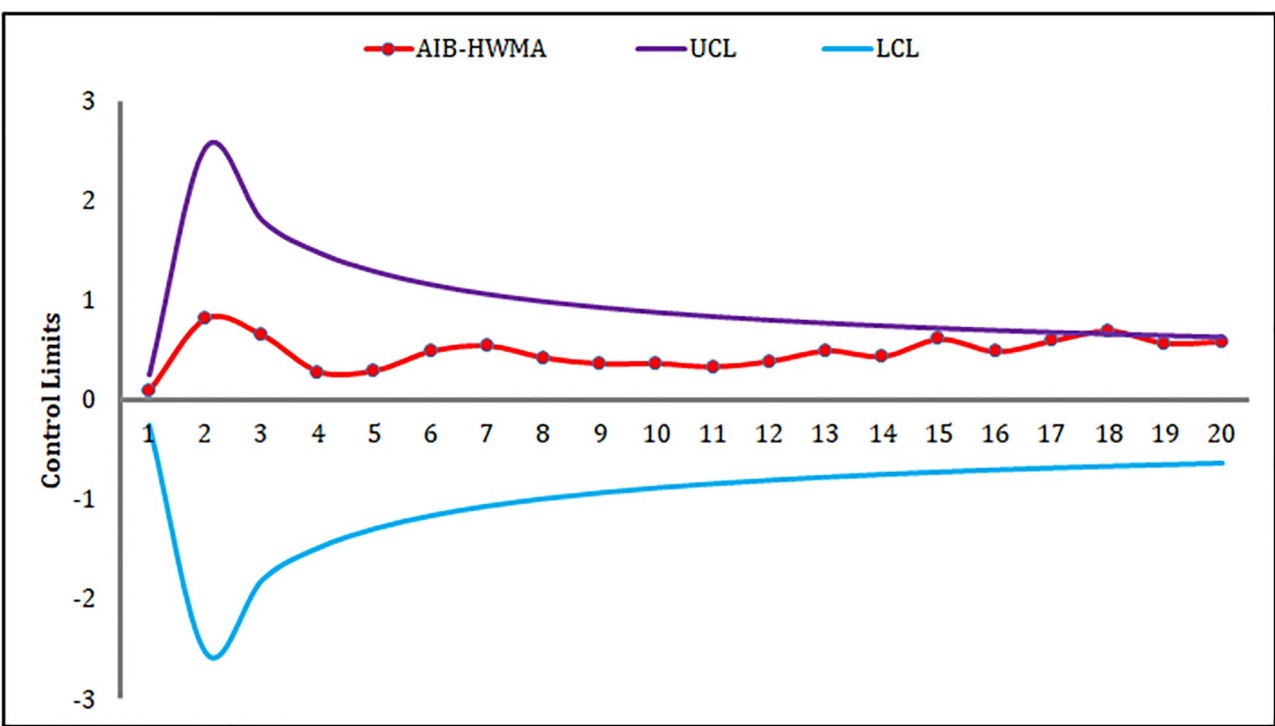

**Fig 4. An application of the HWMA$_{AIB}$ chart.**

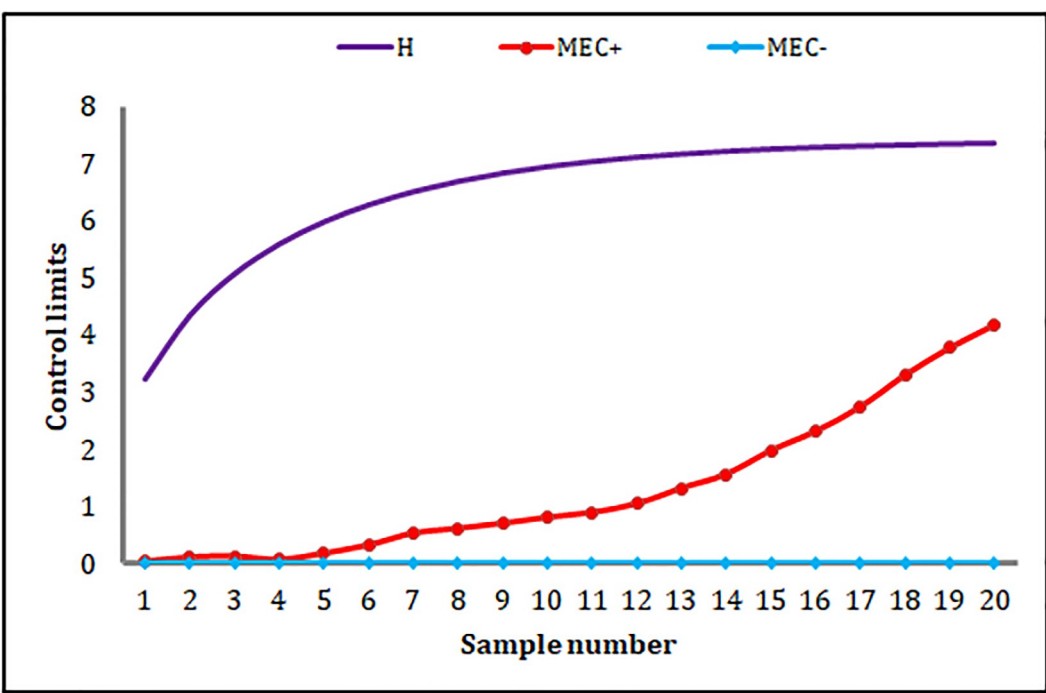

**Fig 5. An application of the MEC$_{AIB}$ chart.**

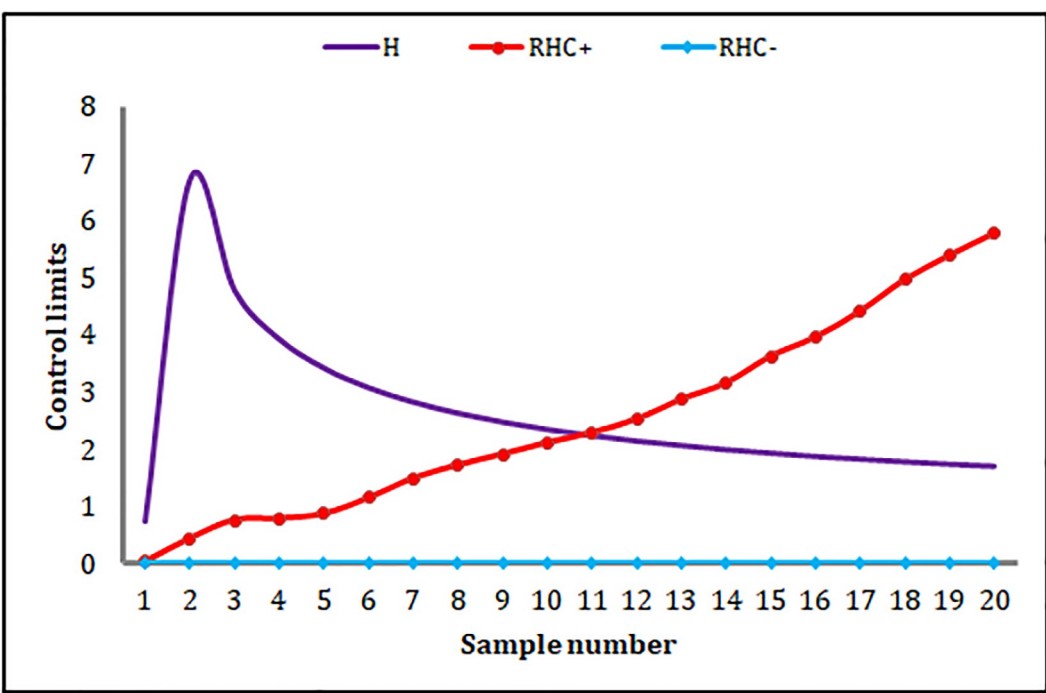

**Fig 6. An application of the MHC$_{AIB}$ chart.**

exceptionally well in detecting shifts over its competitor charts for all selected sets of $\theta$ and $\rho_{zx}$. Also, it is noticed that the choice of the larger value of $\rho_{zx}$ and a smaller value of $\theta$ is effective in enhancing the performance of the MHC$_{AIB}$ chart. An application based on simulated data has also identified the dominance of the MHC$_{AIB}$ chart against the HWMA$_{AIB}$ and MEC$_{AIB}$ charts.

This study can also be extended for dual auxiliary information-based regression estimator for detecting deviations in the process location and dispersion.

## Supporting information

**S1 File.**
(DOCX)

## Author Contributions

**Conceptualization:** Faiza Zubair.

**Formal analysis:** Faiza Zubair.

**Methodology:** Faiza Zubair, Rehan Ahmad Khan Sherwani, Muhammad Abid.

**Supervision:** Rehan Ahmad Khan Sherwani, Muhammad Abid.

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
