## [Decision Letter · Decision Letter 0]

7 Feb 2023

PONE-D-23-00592Enhanced Performance of Mixed HWMA-CUSUM charts using Auxiliary InformationPLOS ONE

Dear Dr. zubair,

Thank you for submitting your manuscript to PLOS ONE. After careful consideration, we feel that it has merit but does not fully meet PLOS ONE’s publication criteria as it currently stands. Therefore, we invite you to submit a revised version of the manuscript that addresses the points raised during the review process.

We look forward to receiving your revised manuscript.

Kind regards,

Robin Haunschild

Academic Editor

PLOS ONE

Journal Requirements:

Additional Editor Comments:

One reviewer was very critical regarding your manuscript. Another reviewer was critical, too. Please be sure to take the reviewers' comments into account and to provide a point-by-point response to each of the comments of the reviewers. Revision of your manuscript does not guarantee acceptance eventually. The outcome of the revision will determine a final decision.

Reviewers' comments:

Reviewer's Responses to Questions

**Comments to the Author**

1. Is the manuscript technically sound, and do the data support the conclusions?

Reviewer #1: Yes

Reviewer #2: Yes

Reviewer #3: No

2. Has the statistical analysis been performed appropriately and rigorously? 

Reviewer #1: Yes

Reviewer #2: Yes

Reviewer #3: No

3. Have the authors made all data underlying the findings in their manuscript fully available?

Reviewer #1: Yes

Reviewer #2: Yes

Reviewer #3: Yes

4. Is the manuscript presented in an intelligible fashion and written in standard English?

Reviewer #1: No

Reviewer #2: Yes

Reviewer #3: No

5. Review Comments to the Author

Reviewer #1: Due to the technological revolution, most processes are equipped with sensors and advanced machines, making them capable of conforming most of the time. Hence, high-quality processes generate quality characteristics and linearly associated information (auxiliary information). Auxiliary information-based monitoring methods are required in the current era of industry 4.0. In this study, a mixed homogeneously weighted moving average (HWMA) cumulative sum (CUSUM) control chart is proposed based on the auxiliary information-based (AIB) regression estimator. A Monte Carlo simulation study is designed to evaluate the performance of proposed control charts in terms of run length metrics such as ARL and SDRL. Moreover, a real-life example is provided to highlight the importance of the stated study. The article is well-written and scientifically sound, and the outcomes of this research are significant and will be helpful for Quality Control readers. I do not have many suggestions due to its current state of the art but before the final acceptance, kindly consider the following points in the revised manuscript.

1. The authors provided a substantial review of the existing studies, but their focus was not on high-quality monitoring methods. Hence, it is suggested to add literature on high-quality process monitoring methods. Moreover, adding recent literature regarding auxiliary information-based control charts and the HWMA chart is highly recommended.

2. It is also suggested to add limitations to the study.

Reviewer #2: It is a good idea. The following comments should be incorporated.

1. Provide the comparison based on steady state along with zero state. Also provide the relevant codes in the appendix section of the paper to verify the results.

2. provide the detailed comparison with some other HWMA types ( DHWMA, THWMA etc.) charts instead of classical CUSUM and EWMA charts.

Reviewer #3: Report

General comments

In this manuscript, the performance of the mixed homogeneously weighted moving average (HWMA) cumulative sum (CUSUM) control chart is improved using the auxiliary information based (AIB) regression estimator, denoted as MHCAIB. The proposed MHCAIB chart is provided. The performance of the proposed MHCAIB is evaluated in terms of the run length properties and the proposed chart is compared with existing AIB charts like CUSUMAIB, EWMAAIB, MECAIB (mixed AIB EWMA-CUSUM), and HWMAAIB. The RL based performance comparisons indicate that the MHCAIB chart achieves relatively better performance to monitor small to moderate shifts over its competitors. The manuscript is not well-written and it is full of mistake. My recommendation is to reject it.

Major comments:

1. The manuscript is not well-written. Authors should consider to use an English language editor to improve the writing.

2. The motivation of this manuscript is very week. You cannot introduce a control chart because others introduced similar charts. Moreover, it is stated in the literature that the HWMA chart performs better for small shifts in the process parameter. Is there any need to combined it with the CUSUM and afterwards use an AIB to improve it again? What do you want to achieve? Did you consider the complexity of the resulting chart?

3. It looks like the authors did not check the formulas properly. There are many mistakes. Below Equation (1) what do y ®_m and t? I think they were used by mistake. Check other equations and derivations as well.

4. Knoth et al. (2021) advised against using the HWMA chart. They stated that the HWMA chart performs worst in steady-state and it is not recommended in practice. The same authors advised against the use of AIB chart in another article. Since both procedures are criticized, do you think that the resulting chart will be acceptable? Elaborate on this.

5. I recommend the investigation of the proposed MHCAIB chart in terms of the conditional expected delay (CED) and compare it to its counterparts considered in this manuscript in terms of the CED properties.

Minor comments:

6. The first time an acronym is used, it must be defined fully- see for instance, RL in the abstract.

7. In the introduction, page 8, add a space between Shewhart and was. Check the entire manuscript to fix similar issues and add space where required.

8. Provide more recent citations

9. Fix the punctuation issues throughout the manuscript.

10. The list of mistakes is very long reason why I did not include it in the report.

6. PLOS authors have the option to publish the peer review history of their article (what does this mean?). If published, this will include your full peer review and any attached files.

Reviewer #1: No

Reviewer #2: No

Reviewer #3: No

---

## [Author Response · Author response to Decision Letter 0]

24 Mar 2023

Response to Reviewers Comments on Article.

1st Reviewers Comments

1. Design is good and study is Significant

2. Some new high quality methods should be added to the literature : “We have added some new studies relevant to high quality methods”

3. Some recent Articles are to be added to the literature based on Auxiliary Information: “Recent studies in year 2023 on the use of HWMA and AIB information” have been added

4. Add limitations: We have added the limitations to the revised article

2nd Reviewer’s Comments

1. Provide Comparisons based on zero state and ready State: with in the recent frame work of the study comparisons based on zero state and ready state cannot be done but we are working on this idea and will present this in some other article. However your suggestion is highly acknowledged.

2. Provide detailed comparison with other HWMA types: in this study we are focusing at the improvements in the design using the auxiliary information so we made comparisons with the existing counterparts which use auxiliary information.

3rd Reviewer’s Comments

1. Consider Use of English language editor to improve language: complying with your kind suggestion we have improved the language of the article and corrected the mistakes.

2. Motivation is weak:

In some industrial and manufacturing processes auxiliary information is also recorded along with the information on the under study variables for different other tasks. This information can be used to improve the design of control chart without imparting any additional financial burden to the entrepreneur. We can use this information for the improvement in design.

Also in high quality monitoring a technique which provides better quality control is required because in the rapidly changing world quality is actually the other name of customers’ satisfaction.

Our proposed chart can be used for high quality monitoring in different industrial applications.

3. Some formulae are not well written: we have made corrections to the formulae and other typing mistakes.

4. Knoth (2021) advised against the use of HWMA and auxiliary information.

Abbas (2022) answered the objection related to use of HWMA for steady states applications. He suggested that a careful choice of HWMA parameter can provide good results even for the steady state. So HWMA can be applied with a careful choice of parameter

Also Knoth’s objection related to the use of auxiliary information is to use it when the auxiliary information is highly correlated. So as per definition of auxiliary information it should be correlated with the study variable, we also suggest the use of the proposed design only when there is a high correlation between the under study variable and the auxiliary variable.

It is also shown that the efficiency of the design improves with the increase in the value of correlation coefficient.

5. Recommended the study in term of Conditional expected delay. Thanks for your kind suggestion. We have started working on this idea and will be presented in future work.

Thanks and Regards.

---

## [Decision Letter · Decision Letter 1]

27 Apr 2023

PONE-D-23-00592R1Enhanced Performance of Mixed HWMA-CUSUM charts using Auxiliary Information

PLOS ONE

Dear Dr. zubair,

Thank you for submitting your manuscript to PLOS ONE. After careful consideration, we feel that it has merit but does not fully meet PLOS ONE’s publication criteria as it currently stands. Therefore, we invite you to submit a revised version of the manuscript that addresses the points raised during the review process.

Please carefully check the points raised by reviewer #3. Please respond if some changes are made in the revised version and also outline the changes in the response.

We look forward to receiving your revised manuscript.

Kind regards,

Robin Haunschild

Academic Editor

PLOS ONE

Additional Editor Comments:

Please carefully check the points raised by reviewer #3. Please respond if some changes are made in the revised version and also outline the changes in the response.

Reviewers' comments:

Reviewer's Responses to Questions

**Comments to the Author**

1. If the authors have adequately addressed your comments raised in a previous round of review and you feel that this manuscript is now acceptable for publication, you may indicate that here to bypass the “Comments to the Author” section, enter your conflict of interest statement in the “Confidential to Editor” section, and submit your "Accept" recommendation.

Reviewer #1: All comments have been addressed

Reviewer #2: All comments have been addressed

Reviewer #3: (No Response)

2. Is the manuscript technically sound, and do the data support the conclusions?

Reviewer #1: Yes

Reviewer #2: Yes

Reviewer #3: No

3. Has the statistical analysis been performed appropriately and rigorously? 

Reviewer #1: Yes

Reviewer #2: Yes

Reviewer #3: No

4. Have the authors made all data underlying the findings in their manuscript fully available?

Reviewer #1: Yes

Reviewer #2: Yes

Reviewer #3: No

5. Is the manuscript presented in an intelligible fashion and written in standard English?

Reviewer #1: Yes

Reviewer #2: Yes

Reviewer #3: No

6. Review Comments to the Author

Reviewer #1: I would like to congratulate the authors for their excellent work in the field of statistical process control. They have made a very detailed revision and convinced me with good arguments. Therefore, I recommend this article for the possible publication.

Reviewer #2: (No Response)

Reviewer #3: The authors did not improve the manuscript according to our suggestions from the previous review.

The motivation of this manuscript is very week. You cannot introduce a control chart because others introduced similar charts. Moreover, it is stated in the literature that the HWMA chart performs better for small shifts in the process parameter. Is there any need to combined it with the CUSUM and afterwards use an AIB to improve it again? What do you want to achieve? Did you consider the complexity of the resulting chart?

It looks like the authors did not check the formulas properly. There are many mistakes. Below Equation (1) what do y ®_m and t? I think they were used by mistake. Check other equations and derivations as well.

Knoth et al. (2021) advised against using the HWMA chart. They stated that the HWMA chart performs worst in steady-state and it is not recommended in practice. The same authors advised against the use of AIB chart in another article. Since both procedures are criticized, do you think that the resulting chart will be acceptable? Elaborate on this.

I recommend the investigation of the proposed MHCAIB chart in terms of the conditional expected delay (CED) and compare it to its counterparts considered in this manuscript in terms of the CED properties.

7. PLOS authors have the option to publish the peer review history of their article (what does this mean?). If published, this will include your full peer review and any attached files.

Reviewer #1: No

Reviewer #2: No

Reviewer #3: No

---

## [Author Response · Author response to Decision Letter 1]

16 Jun 2023

Reply to Reviewer’s Comments.

Manuscript Title: Enhanced Performance of Mixed HWMA-CUSUM charts using Auxiliary Information

We are thankful to the reviewers and Editor(s) for their valuable comments and suggestions, which helped us improve the manuscript.

Reviewer 1 and Reviewer 2 have already given their acceptance and no questions were raised in the second review

3rd Reviewer’s Comments and Replies

Respected Reviewer

Following are the comments and the possible replies from the author(s)

1. Consider Use of English language editor to improve language: 

Reply: Complying with your kind suggestion, we have improved the language of the article and corrected the mistakes. Grammatical errors have been removed, equations are corrected, and errors are identified and rectified in the manuscript.

2. The motivation of this manuscript is very week. You cannot introduce a control chart because others introduced similar charts. Moreover, it is stated in the literature that the HWMA chart performs better for small shifts in the process parameter. Is there any need to combined it with the CUSUM and afterwards use an AIB to improve it again? What do you want to achieve? Did you consider the complexity of the resulting chart?

Reply: HWMA performs better than other charts for small shifts; combining it with the CUSUM chart improves its performance for both larger and smaller shifts. Auxiliary variable further improves the results. As in the recent age of development, improvements in quality assurance techniques are the need of the hour. In context, we have developed a new chart showing visible improvement in detecting shifts. Even a minute change and deviation in the quality can be a big hurdle in many industrial processes like lifesaving drugs, substrate manufacturing, missile equipment, etc. In some industrial and manufacturing processes, the auxiliary information is also recorded along with the information on the under-study variables for different other tasks. This information can be used to improve the control chart design without imparting any additional financial burden to the entrepreneur. We can use this information for the improvement of design. Our proposed chart is shown to have improved results compared to its counterparts and is useful for high-quality monitoring in different industrial applications. (see section 1..introduction.paragraph 2).

3. It looks like the authors did not check the formulas properly. There are many mistakes. Below Equation (1), what do y ®_m and t? I think they were used by mistake. Check other equations and derivations as well. 

Reply: We have made corrections to the formulae and other typing mistakes.

4. Knoth et al. (2021) advised against using the HWMA chart. They stated that the HWMA chart performs worst in steady-state and it is not recommended in practice. The same authors advised against the use of AIB chart in another article. Since both procedures are criticized, do you think that the resulting chart will be acceptable? Elaborate on this. Reply: Since the objection has been raised, different scholars have answered to this objection. Abbas (2022) answered the objection to using HWMA for steady-state applications. He suggested carefully choosing HWMA parameters can provide good results even for the steady state. So HWMA can be applied with a careful choice of parameter.

Also, Knoth’s objection to using auxiliary information is to use it when it is highly correlated. So as per the definition of auxiliary information, it should be correlated with the study variable; we also suggest using the proposed design only when there is a high correlation between the under-study variable and the auxiliary variable.

It is also shown that the efficiency of the design improves with the increase in the correlation coefficient value.

5. I recommend the investigation of the proposed MHCAIB chart in terms of the conditional expected delay (CED) and compare it to its counterparts considered in this manuscript in terms of the CED properties. 

Reply: Thanks for your kind suggestion. We have started working on this idea, and it will be presented in future work.

Thanking you in anticipation.

Regards

---

## [Decision Letter · Decision Letter 2]

15 Aug 2023

Enhanced Performance of Mixed HWMA-CUSUM charts using Auxiliary Information

PONE-D-23-00592R2

Dear Dr. zubair,

We’re pleased to inform you that your manuscript has been judged scientifically suitable for publication and will be formally accepted for publication once it meets all outstanding technical requirements.

Kind regards,

Robin Haunschild

Academic Editor

PLOS ONE

Additional Editor Comments (optional):

Please consider if it is useful to include some of the following thoughts of another reviewer in the manuscript in one of the following stages:

The reviewer mentioned the criticism on HWMA charts. I believe there are several short production runs with low-volume manufacturing environments where the quality of characteristics is monitored (Berardinelli, 2017). The shifts in such processes, if any, occur at or close to the start. The current study targets such processes where early change point detection is desired for shifts that occur close to the beginning of the process. Accordingly, the algorithm of zero-state ARL computation is used where the possible change point is at i=1. The current proposal is not targeted towards long production runs where one may refer to steady-state ARL results. Hence, it has been shown by Knoth et al. (2021) that HWMA does not work for long production runs, while N. Abbas (2018) showed HWMA efficacy under short production runs. Moreover, after the criticism by Knoth et al. (2021), Riaz et al. (2022) proposed a revised version of the HWMA chart, which can work for both short and long production runs.

Further, combining the HWMA with the CUSUM chart enhanced the performance for larger and smaller shifts presented in this study. The similar studies on combining different charts with CUSUM chart, one may see (Nasir Abbas et al., 2018; Nasir Abbas et al., 2013; Ajadi & Riaz, 2017; Ajadi et al., 2016; Riaz et al., 2019).

Further, Saleh et al. (2023) and Saleh et al. (2022) criticize the Auxiliary Information Based (AIB) control charts. However, to answer the objections, one may see (Nasir Abbas et al., 2022).

Finally, conditional expected delay (CED) is used to measure performance when examining a steady-state ARL study. It is clear that the authors are comparing the charts based on zero-state ARL computations. Hence, I believe there is no need to find and report CED measures.

I found no problem in combining the HWMA chart with the CUSUM chart. I believe the authors showed that this combination had improved the detection ability for more extensive and smaller shifts.

References:

Abbas, N. (2018). Homogeneously weighted moving average control chart with an application in substrate manufacturing process. Computers & Industrial Engineering, 120, 460-470. doi:https://doi.org/10.1016/j.cie.2018.05.009

Abbas, N., Ahmad, S., & Riaz, M. (2022). Reintegration of auxiliary information based control charts. Computers & Industrial Engineering, 171, 108479. doi:https://doi.org/10.1016/j.cie.2022.108479

Abbas, N., Raji, I. A., Riaz, M., & Al-Ghamdi, K. (2018). On Designing Mixed EWMA Dual-CUSUM Chart With Applications in Petro-Chemical Industry. IEEE Access, 6, 78931-78946.

Abbas, N., Riaz, M., & Does, R. J. (2013). Mixed exponentially weighted moving average–cumulative sum charts for process monitoring. Quality and Reliability Engineering International, 29(3), 345-356.

Ajadi, J. O., & Riaz, M. (2017). Mixed multivariate EWMA-CUSUM control charts for an improved process monitoring. Communications in Statistics - Theory and Methods, 46. doi:10.1080/03610926.2016.1139132

Ajadi, J. O., Riaz, M., & Al-Ghamdi, K. (2016). On increasing the sensitivity of mixed EWMA–CUSUM control charts for location parameter. Journal of Applied Statistics, 43. doi:10.1080/02664763.2015.1094453

C. Berardinelli, Short-Run Statistical Process Control Techniques. https://www.isixsigma.com/control-charts/short-run-statistical-process-control-techniques/, 2017 (accessed 06 April 2023).

Knoth, S., Tercero-Gómez, V. G., Khakifirooz, M., & Woodall, W. H. (2021). The impracticality of homogeneously weighted moving average and progressive mean control chart approaches. Quality and Reliability Engineering International, 37(8), 3779-3794. doi:https://doi.org/10.1002/qre.2950

Riaz, M., Ahmad, S., Mahmood, T., & Abbas, N. (2022). On Reassessment of the HWMA Chart for Process Monitoring. Processes, 10(6), 1129.

Riaz, M., Ajad, J. O., Mahmood, T., & Abbasi, S. A. (2019). Multivariate Mixed EWMA-CUSUM Control Chart for Monitoring the Process Variance-Covariance Matrix. IEEE Access, 7, 100174-100186. doi:10.1109/access.2019.2928637

Saleh, N. A., Mahmoud, M. A., Woodall, W. H., & Knoth, S. (2022). Guest editorial on auxiliary information based monitoring. Quality and Reliability Engineering International, 38(5), 2255-2255.

Saleh, N. A., Mahmoud, M. A., Woodall, W. H., & Knoth, S. (2023). A review and critique of auxiliary information-based process monitoring methods. Quality Technology & Quantitative Management, 20(1), 1-20.

Reviewers' comments:

Reviewer's Responses to Questions

**Comments to the Author**

1. If the authors have adequately addressed your comments raised in a previous round of review and you feel that this manuscript is now acceptable for publication, you may indicate that here to bypass the “Comments to the Author” section, enter your conflict of interest statement in the “Confidential to Editor” section, and submit your "Accept" recommendation.

Reviewer #1: All comments have been addressed

Reviewer #2: All comments have been addressed

2. Is the manuscript technically sound, and do the data support the conclusions?

Reviewer #1: Yes

Reviewer #2: Yes

3. Has the statistical analysis been performed appropriately and rigorously? 

Reviewer #1: Yes

Reviewer #2: Yes

4. Have the authors made all data underlying the findings in their manuscript fully available?

Reviewer #1: Yes

Reviewer #2: Yes

5. Is the manuscript presented in an intelligible fashion and written in standard English?

Reviewer #1: Yes

Reviewer #2: Yes

6. Review Comments to the Author

Reviewer #1: I would like to congratulate the authors for their excellent work in the field of statistical process control. They have made a very detailed revision and convinced me with good arguments. Therefore, I recommend this article for the possible publication.

Reviewer #2: (No Response)

7. PLOS authors have the option to publish the peer review history of their article (what does this mean?). If published, this will include your full peer review and any attached files.

Reviewer #1: **Yes: **Tahir Mahmood

Reviewer #2: No

---

## [Editor Report · Acceptance letter]

7 Sep 2023

PONE-D-23-00592R2 

Enhanced Performance of Mixed HWMA-CUSUM charts using Auxiliary Information 

Dear Dr. zubair:

I'm pleased to inform you that your manuscript has been deemed suitable for publication in PLOS ONE. Congratulations! Your manuscript is now with our production department. 

Kind regards, 

on behalf of

Dr. Robin Haunschild 

Academic Editor

PLOS ONE